# "Dysregulated not deficit": A qualitative study on symptomatology of ADHD in young adults

Callie M. Ginapp[1¤a]*, Norman R. Greenberg[1¤b], Grace MacDonald-Gagnon[2¤c], Gustavo A. Angarita[2,3], Krysten W. Bold[2,4], Marc N. Potenza[2,3,4,5,6,7,8]*

1 Yale University School of Medicine, New Haven, Connecticut, United States of America, 2 Department of Psychiatry, Yale School of Medicine, New Haven, Connecticut, United States of America, 3 Connecticut Mental Health Center, New Haven, Connecticut, United States of America, 4 Yale Cancer Center, New Haven, Connecticut, United States of America, 5 Connecticut Council on Problem Gambling, Wethersfield, Connecticut, United States of America, 6 Child Study Center, Yale School of Medicine, New Haven, Connecticut, United States of America, 7 Department of Neuroscience, Yale University, New Haven, Connecticut, United States of America, 8 Wu Tsai Institute, Yale University, New Haven, Connecticut, United States of America

¤a Current address: Beth Israel Deaconess Medical Center, Department of Psychiatry, Boston, Massachusetts, United States of America
¤b Current address: Weill Cornell Medicine, Department of Psychiatry, New York, New York, United States of America
¤c Current address: Department of Psychology, University of Illinois Chicago, Chicago, Illinois, United States of America
* cginapp@bidmc.harvard.edu (CMG); marc.potenza@yale.edu (MNP)

**Data Availability Statement:** All relevant data are within the manuscript and its Supporting information files.

## Abstract

### Objective

Attention-deficit/hyperactivity disorder (ADHD) is a common condition that often persists into adulthood, although data suggest that the current diagnostic criteria may not represent how the condition presents in adults. We aimed to use qualitative methods to better understand ADHD symptomatology in young adults, especially regarding attentional and emotional dysregulation.

### Methods

Nine focus groups involving young adults (aged 18–35 years; N = 43; 84% female; 86% US and Canada) with diagnoses of ADHD were conducted. Participants were asked about their perceptions of the current diagnostic criteria and how their symptoms have presented and changed over time. Data were analyzed using an interpretive phenomenological analysis framework.

### Results

Most participants reported that the diagnostic criteria did not accurately capture their experiences with ADHD. They reported struggling with attention dysregulation, including hyperfocusing, and emotional dysregulation, including rejection-sensitive dysphoria. Many participants believed that their changing environments and behavioral adaptations influenced how their symptoms presented into adulthood.

**Funding:** This work was supported by the Yale School of Medicine Office of Student Research One Year Fellowship and the NIDA K12 DA000167 grant. The funders had no role in study design, data collection and analysis, decision to publish, or preparation of the manuscript.

**Competing interests:** The authors report no conflicts of interest with the topic of this work. Marc N. Potenza has consulted for and advised Opiant Pharmaceuticals, Idorsia Pharmaceuticals, Baria-Tek, AXA, Game Day Data and the Addiction Policy Forum; has been involved in a patent application with Yale University and Novartis; has received research support from the Mohegan Sun Casino and Connecticut Council on Problem Gambling; has participated in surveys, mailings or telephone consultations related to drug addiction, impulse control disorders or other health topics; and has consulted for law offices and gambling entities on issues related to impulse control or addictive disorders. This does not alter our adherence to PLOS ONE policies on sharing data and materials. The other authors report no disclosures.

## Conclusion

Current diagnostic criteria for ADHD may not capture the range of symptoms present in young adults. More research is needed to characterize attentional and emotional dysregulation in this population.

## Introduction

Attention-deficit/hyperactivity disorder (ADHD) is classified as a neurodevelopmental disorder in the fifth edition of the Diagnostic and Statistical Manual (DSM-5) [1]. ADHD has historically been considered a disorder of childhood, but data suggest between a 3%-7% prevalence of ADHD in adults globally [2, 3]. When studies utilize recommended practices to reassess adults including an age-adjusted symptom threshold, ADHD persistence estimates are between 40% and 50% in adults [4].

First introduced into the DSM-II in 1968 as hyperkinetic reaction of childhood and later being termed ADHD, the disorder continues to be diagnosed based on symptoms that reflect presentation in children, particularly males [1]. These features include inattention, hyperactivity, and impulsiveness [1]. Diagnostic criteria have changed over time to better encompass considerations of experiences of adults by lessening the number of necessary symptoms present in adulthood, but the symptoms themselves are still based in how the condition presents at a young age and in males [1, 5]. Data suggest that childhood symptoms may not be encompassing of the symptomatology of adults, especially with regard to attentional and emotional regulation [5]. A diagnosis of ADHD in adulthood often involves a combination of self-reported scales, interviews, and longitudinal family histories [6], although there are currently no standardized guidelines for diagnosing ADHD in adults. ADHD is underdiagnosed in adults [7]; a recent study in Italy found that the median duration of untreated illness for adults diagnosed with ADHD was 17 years and is longer for those with the inattentive subtype in childhood [8]. Rates of underdiagnosis and delayed diagnosis are higher among females than males, and females are underrepresented in ADHD research [9]. This may be in part due to females being more likely to experience the inattentive sub-type instead of the hyperactive/impulsive sub-type when compared to males [10]. Delays in diagnosis of ADHD may contribute importantly to functional impairments across many domains including academic difficulties, unemployment, interpersonal conflict, and high rates of substance abuse [6].

Problems with attentional regulation may include difficulties redirecting attention away from certain tasks, also known as hyperfocusing. Tendencies of individuals with ADHD to hyperfocus have been previously described [11], and recent studies suggest that individuals with ADHD report longer and more frequent episodes of hyperfocusing compared to individuals without ADHD [12, 13]. One study reported that people with and without ADHD did not differ in overall frequency of reported hyperfocusing, but those with ADHD were less likely to hyperfocus during educational and social activities [14]. These data suggest that adults with ADHD may not have an overarching deficit of attention as the current diagnostic criteria suggest, but rather periodic episodes of increased attention and differences in directing and orientating attention.

Despite emotional dysregulation being changed from a formal criterion of ADHD to an associated feature in the DSM-III [15], emotional dysregulation has been described among adults with ADHD [16–19] and there remains debate regarding whether emotional dysregulation constitutes a core feature of ADHD [20]. Symptoms of emotional dysregulation in ADHD may include emotional lability resulting in interpersonal conflict [20]. Young adults with

ADHD have been reported to be twice as likely to experience emotional dysregulation, and emotional dysregulation has been linked to lower quality of life independent of traditional ADHD symptomatology [21]. Emotional dysregulation may be particularly relevant to females with ADHD.

Together, these findings suggest ADHD may present differently in adults than the current diagnostic criteria suggest. Qualitative research may provide insight into the lived experiences of young adults with ADHD which could inform refinements in future diagnostic criteria. The qualitative approach seeks to generate new hypotheses which can then be further explored quantitatively. This study aims to understand how young adults with ADHD perceive the current diagnostic criteria and their attentional and emotional symptoms and how their symptoms have changed overtime.

## Methods

### Recruitment

Participants were recruited from online sources including online communities for people with ADHD such as Facebook and Reddit with permission from the moderators of these spaces and on the Children and Adults with ADD (CHADD) advocacy group website.

Interested participants were first directed to a Qualtrics survey that provided information on the purpose of the study and then they had an opportunity to complete an initial screening survey to determine eligibility. Inclusion criteria consisted of being aged 18–35 years, inclusively, reporting being diagnosed with ADHD by a clinician, and scoring greater than or equal to 23 on the Adult ADHD Self-Report Scale, reflecting on symptoms over the life course [22]. This study was based out of the United States, although participants could reside in any country.

### Procedure

The following procedure was approved by the Yale University Institutional Review Board (2000031327), and participants provided verbal consent before participation. After completion of the screening survey, invitations were extended via email to potential participants to schedule a fifteen-minute meeting, during which participants provided verbal consent and completed a demographic questionnaire. Verbal consent was utilized in this remote study as to minimize the amount of paperwork participants had to complete and return. The script used during consent meetings (S1 Appendix) and the informed consent document distributed to participants (S2 Appendix) can be found in the Appendices. Participants were informed of the purpose of the study, study logistics including video and audio recording of focus groups, and potential risks of participation such as potential loss of confidentiality and emotional discomfort.

There were 146 completed responses to the screening survey, and 61 of these potential participants responded to the follow-up emails. Of these, 51 potential participants completed informed consent for the study and 43 ultimately took part in a focus group, all of whom participated in their group. Reasons for not completing the study after responding were largely participant non-response without explanation, although some participants expressed that life circumstances had come up that prevented them from continuing their involvement. An additional 23 people completed the screening survey after the focus groups had reached saturation and were turned away from the study.

In addition to meeting criteria on the Adult ADHD Self-Report Scale, participants were asked to either provide documentation of diagnosis or complete a release of information to confirm diagnosis with their medical provider. The Connecticut State Prescription Monitoring

program, which supports 39 states, was also checked with participant permission to verify if they were currently being prescribed stimulants as a proxy for ADHD diagnosis. Twenty-seven (63%) of the participants had a confirmation of diagnosis by either method. Participants who did not provide documentation were not excluded from the study. There were no instances of discrepancies between providers' and participants' reports of ADHD diagnoses, and all participants scored above threshold on the Adult ADHD Self-Report Scale.

Nine focus groups over November and December 2021 were conducted with three to six participants per group. All focus groups took place over Zoom and were video- and audio-recorded with participant consent. Participants were compensated with a $15 e-gift card after completion of the focus group. The focus groups lasted approximately one hour and were led by one facilitator with an additional member of the research team present for logistical concerns. Focus groups were semi-structured and facilitated by a discussion guide. The guide included 1) questions about how the current definition of ADHD represented participants' lived experiences, 2) symptomatology of ADHD such as attention and emotional dysregulation, and 3) how these ADHD symptoms changed over their lives. In general, the first opening questions were sometimes more closed (for example, "do you feel. . ." and "do you experience. . ."). These were typically followed by more open-ended questions (for example, beginning with "how. . ."). The initial directed questions were asked to gauge agreement or disagreement among the group and to understand whether participants had specific experiences. Subsequently, the more open-ended follow-up questions were asked to better understand participant experiences. The discussion guide can be found in S3 Appendix.

Participants were also asked about community building in online spaces and interpersonal relationships, these results have been published separately [23]. After completion of the focus groups, participants were provided with a summary document of main themes from the groups. While alternate approaches were considered, focus groups were considered optimal. For example, while analysis of material from online forums potentially would have been feasible, we believe that the focus groups are more transparent regarding the research nature of the work. That is, when contributing to online forums, participants were not necessarily consenting to have their input used for research purposes, whereas for the focus groups, they were.

## Theoretical framework and role of the researcher

An Interpretive Phenomenological Analysis (IPA) approach was used for this study as it employs the double hermeneutic (two-way-relationship understanding) regarding participants' understanding of their lived experiences and the researchers' interpretation of participants' construction of their experiences [24]. We focused the analysis on the level of the individual's subjective lived experience. Although IPA traditionally utilizes individual interviews [25], focus groups were employed in this study, and we employed an IPA approach that has been adapted for use in focus groups [26]. An IPA approach adapted for focus groups was chosen because of the research focus on participants' subjective lived experiences and how people ascribe meaning to these lived experiences. We performed analysis on the individual level before performing wider group analysis, consistent with the approach of using IPA for analysis of data from focus groups [26]. Focus groups were used in place of individual interviews because being in groups of people with similar experiences can promote participant disclosure of potentially stigmatizing topics [27], and focus groups have been previously used in qualitive studies of young adults with ADHD for this purpose [28]. Focus groups also allow for agreements and disagreements across topics to be readily captured.

The first author facilitated the focus groups, and the first and second author conducted the data analysis. These authors have unique experiences that supported the study goals. The first

author is a medical student who is diagnosed with ADHD, is involved with online communities for adults with ADHD, and has conducted a review on adult ADHD qualitative research. The second author is a medical student whose previous research has focused on impulsivity and behavioral addictions and has several family members with ADHD. This insight into the ADHD community allowed for knowledge about where to recruit participants and contributed to the development of the discussion guide. During the individual informed consent meetings, many participants asked the lead researcher's personal motivation for conducting the study, and some directly asked if the researcher had ADHD. When these questions were asked, the researcher disclosed their ADHD diagnosis which was considered important in establishing rapport. The researcher was familiar with the vocabulary used by participants and self-conceptualization of adult ADHD as described in online communities. This allowed for informed follow-up questions during the focus groups and provided context when analyzing the data. Potential biases include the researchers' preconceived perceptions of ADHD based on their personal experiences and potential projection of these experiences onto participants. To mitigate these biases, the lead researcher did not discuss their experiences with ADHD with participants, journaled to reflect on their experiences and the possible biases they created, and remained cognizant of these biases while conducting data analyses. Transcripts were iteratively discussed and analyzed by both researchers to help limit bias in interpreting statements by participants.

## Data analysis

Recordings were transcribed verbatim and checked for accuracy by a member of the research team. Transcripts were uploaded to NVivo 12. After reading the transcripts, two members of the research team mutually developed a preliminary code book using an IPA approach [26]. Half of the transcripts were independently coded by two reviewers. Discrepancies were resolved by mutual agreement, and the code book was interactively updated. The remaining transcripts were coded with the updated code book by the lead researcher. Codes were developed inductively with focus on the individual experience. In developing themes, attention was paid to how participants were represented by the group-level themes. Participant agreement and disagreement across themes and focus groups were captured by creation of subthemes (i.e., subthemes agree and disagree under the theme of opinions on diagnostic criteria). There was considerable overlap in themes across focus groups, although stand-alone themes are also reported. All quotations are provided in the supplementary appendix with additional subheadings to provide additional context (S4 Appendix). In describing results, we have at times used singular versions of them/their descriptors, particularly as 16% of participants were non-binary.

## Results

### Participant characteristics

Among the 43 participants, the median age was 29 years and median age of ADHD diagnosis was 22 years (range 5–34 years; Table 1). The median time between diagnosis and entering this study was three years and the mean was seven years. Participants were 84% female, 72% White, 14% Asian, 9% Black, and 5% Hispanic/Latino. Seventy-two percent of participants were from the United States, fourteen percent were from Canada, and the remaining were from Australia, Suriname, the Czech Republic, and the United Kingdom. Forty-four percent were currently students, and 88% had at least some college. The most common ADHD subtype was inattentive, followed by combined, with only 9% having been diagnosed as predominately hyperactive/impulsive. Nearly a quarter of participants either did not know their subtype or

**Table 1. Participant demographics; N = 43.**

|  |  | Number | Percent |
|---|---|---|---|
| Age | 18–24 | 11 | 26% |
|  | 25–29 | 14 | 33% |
|  | 30–35 | 18 | 42% |
| Age at diagnosis | <18 | 11 | 26% |
|  | ≥18 | 32 | 74% |
| Sex | Male | 7 | 16% |
|  | Female | 36 | 84% |
| Gender | Male | 8 | 19% |
|  | Female | 28 | 65% |
|  | Non-binary/gender queer | 7 | 16% |
| Race | White | 31 | 72% |
|  | Black/African American | 4 | 9% |
|  | Asian | 6 | 14% |
|  | Other | 2 | 5% |
| Ethnicity | Hispanic/Latino | 2 | 5% |
| Country | United States | 31 | 72% |
|  | Canada | 6 | 14% |
|  | Other | 7 | 16% |
| Education | High school/GED | 5 | 12% |
|  | Some college | 15 | 35% |
|  | Associate's degree | 2 | 5% |
|  | Bachelor's degree | 13 | 30% |
|  | Graduate degree | 7 | 16% |
| Employment | Student | 19 | 44% |
|  | Employed full time | 23 | 53% |
|  | Employed part time | 8 | 19% |
|  | Unemployed | 3 | 7% |
| ADHD sub-type | Inattentive | 18 | 42% |
|  | Hyperactive/impulsive | 4 | 9% |
|  | Combined | 12 | 28% |
|  | Unknown | 9 | 21% |
| Stimulants | Current | 29 | 67% |
|  | Previous | 7 | 16% |
|  | Never | 7 | 16% |
| Therapy | Current | 26 | 60% |
|  | Previous | 3 | 7% |
|  | Never | 14 | 33% |
| Comorbidities | Depression | 22 | 51% |
|  | Anxiety | 21 | 49% |
|  | Autism | 4 | 9% |
|  | None | 11 | 26% |
| ADHD social media involvement | Facebook | 31 | 74% |
|  | TikTok | 17 | 40% |
|  | Instagram | 14 | 33% |
|  | Reddit | 12 | 28% |
|  | Twitter | 6 | 14% |
|  | None | 3 | 7% |

*(Continued)*

**Table 1.** (Continued)

| | | Number | Percent |
|---|---|---|---|
| Recruited from | Facebook | 31 | 72% |
| | CHADD | 10 | 23% |
| | Reddit | 2 | 5% |

were diagnosed before subtypes were introduced. Eighty-four percent of participants had been prescribed stimulant medication, a common pharmacotherapy treatment for ADHD, with nearly two-thirds reporting current stimulant use. Seventy-four percent of participants had been diagnosed with at least one other psychiatric condition, most commonly anxiety and depression. Over half of participants were currently in psychotherapy for any reason. Most participants were recruited from Facebook, followed by CHADD and then then Reddit. Ninety-three percent of participants were involved with ADHD communities on social media, most commonly Facebook followed by TikTok, Instagram, Reddit, and Twitter.

## Diagnosis

**Diagnostic criteria.** When asked if they felt the current diagnostic criteria for ADHD described their lived experiences, participants in some focus groups agreed with the criteria. However, these agreements were generally not further elaborated (Table 2). Most participants had apprehensions regarding the current definition of ADHD and felt that the symptoms of inattention, impulsivity, and hyperactivity did not capture the full experience of living with ADHD. Many participants did not experience hyperactivity and therefore were hesitant about

**Table 2. Representative quotes on perceptions of diagnosis.**

| Diagnostic criteria | Agree | I'd say it's accurate from my experiences.<br>Me too.<br>Likewise. |
|---|---|---|
| | Disagree | I find that the challenge of the definition of ADHD is that it's limited. . . . impulsivity and hyperactivity are accurate but not comprehensive. So, I think that it's really common for people, including myself, to receive a diagnosis to think that that's all ADHD is. . . I feel like executive dysfunction is more comprehensive. |
| | | I feel like the word inappropriate carries a really negative connotation. . . I feel like the word inappropriate is kind of a loaded term to put into a medical diagnosis. . .. Right, like inappropriate on whose standards. |
| | | I would say developmentally inappropriateness is like a delay or advancement, but it usually doesn't change or specifically catch up to other developmental delays. It's not like it's behind or ahead of, it's separate like it's a parallel, not on the same track. |
| Misdiagnosis | | I wasn't diagnosed until I was 22, 23, something like that. And so, my whole childhood I had these symptoms that were misdiagnosed as anxiety and depression and all of these different things.. . . I went without getting any real relief. |
| | | I was diagnosed with depression and anxiety quite young but it's due to the ADHD because I notice when I take Ritalin. . .I don't really need them [antidepressants]. When I take Ritalin the stuff that's making me anxious or feeling depressed, it's not there anymore. |
| Wished diagnosed sooner | | Thinking back on it, I'm a social worker now, I've studied how all of this presents. There were a million red flags. I needed help sooner than seventeen. |
| | | All the signs were there, why wasn't it caught?. . . I'm 34 and I was just diagnosed in August so it's all brand new. All these puzzle pieces falling into place. All everything in my life leading up to now just makes sense. |

it being considered a core feature of ADHD. The term executive dysfunction was seen as more comprehensive in describing their symptoms.

The term developmentally inappropriate was problematic for many participants. It was viewed as stigmatizing, infantilizing, and inaccurate, especially for how ADHD presents in adults. Developmentally inappropriate behavior was sometimes interpreted as a delay, which participants did not view as an accurate representation of the condition. Rather, they viewed ADHD as separate path as opposed to a delay in ultimately following the same developmental pathway.

**Misdiagnosis.**   Many participants reported having been originally misdiagnosed with other mental health disorders such as anxiety and depression before receiving their ADHD diagnosis. Participants attributed these misdiagnoses to physician's considering their anxious or depressive symptoms as being representative of a primary mood or anxiety disorder instead of exploring other situational or underlying factors generating these emotions. Difficulties in maintaining societal responsibilities due to ADHD symptoms often manifested in feelings of anxiety or depression. Often once individuals were treated for ADHD, they reported that their anxious or depressive symptoms typically subsided.

**Timing of diagnoses.**   Many participants regretted that they had not been diagnosed as children as they grew up without treatment and without explanations for their symptoms. Participants commonly mentioned feelings of disbelief that their severe symptoms had been overlooked and relief after having finally been diagnosed.

## Symptomatology

**Attention.**   *"Dysregulated not deficit"*. No participants reported a universal deficit of attention; rather, participants conceptualized their attention as dysregulated (Table 3). This dysregulation was experienced as intermittent abundance of attention and was described as hyperfocusing, or having difficulties redirecting attention away from interesting tasks. Participants reported attending to tasks for multiple hours with high levels of concentration. This experience was at times seen as helpful to accomplishing work, especially academic tasks.

Although helpful at times, hyperfocusing came with its own set of problems. Participants reported feeling unable to actively choose both when to hyperfocus and on what to hyperfocus, often resulting in hours spent on activities without feeling like they decided to do so. In addition to not prioritizing work that often needed to be done, hyperfocusing came at the detriment of basic life tasks such as eating and sleeping. Time blindness, or difficulty perceiving how much time had passed, was reported during episodes of hyperfocusing. Participants felt they could not discontinue hyperfocusing of their own accord; when interrupted from an episode by others, they often became irritable, especially to physical touch.

*Facilitators and barriers to focusing.* Topics of intrinsic interest were unanimously reported as easier on which to focus; boring topics were viewed as nearly impossible to which to attend. Tasks that were novel or engaging were also seen as easier for maintaining focus. Deadlines had a strong impact on attention, leaving many to procrastinate working on assignments until the "last minute" as it was then easier to focus.

The physical external environment modulated attention, with the "right" environment, one free from clutter or distractions, promoting productivity. Sound often played a critical role in shaping the environment, with select ambient sound helping with focus and particular noises such as people talking or typing and air conditioners being the most troublesome distractions. Some participants reported changes in ability to focus with the time of day and how rested they felt. Strategies to help improve focus included fidgeting and, in one focus group, consuming marijuana, caffeine or other stimulants (see below).

**Table 3. Representative quotes on attention.**

| | | |
|---|---|---|
| "Dysregulated not deficit" | | I think that deficit might not be the right word because I have an abundance of attention at times, it's just that I can't direct it. I don't get to choose what I pay attention to. |
| Benefits of hyperfocusing | | I definitely experience hyperfocus and it's like my superpower when it's good. . . I can produce a lot of work in a very short of time and it's going to be high quality. |
| Drawbacks of hyperfocusing | Neglecting other tasks | I'm not going to stop doing that thing. It doesn't matter if I have responsibilities, they have to wait. And unfortunately, it's rarely my responsibilities that trigger that hyperfocus. |
| | | It's very inconvenient but in that case it does kind of impair your functioning because I was set out to study and instead, I colored something for three hours. |
| | | There've been countless times I forget to eat, I won't hydrate, I won't go to the bathroom. |
| | Time blindness | I'm like oh, I'll just look at this for five minutes and then 45 minutes go by or an hour, and why did I research some author, why did I research whatever? It's an hour and a half or more digging into something you weren't supposed to. |
| | | I just get lost you know, six hours black out. |
| | Irritability | I get super furious too when I'm touched or interrupted during hyperfocus because you're so zoned into what you're doing. |
| Facilitators and barriers to focusing | Facilitators | When something stimulates me or is interesting to me then yeah, I'm not going to stop doing that thing. |
| | | People with ADHD have a novelty based nervous system. . . .Something new or interesting, or moving my focus to different things is easier. |
| | | If there's a time crunch, if there's pressure, I would do something for hours on end. . . I think pressure really mobilizes that hyperfocus. |
| | Barriers | Loud noises and those distractions, other people can just block them out and they just get their work done or they concentrate on something. I absolutely can't. Interestingly though, I don't like silence, so I need to have a little bit of background noise, a little bit of light music that helps my brain. |
| | | I find that depending on the environment I can be very easily distracted or more easily focused or hyperfocused. Especially if it's a very cluttered environment which I often find myself in. |
| Difficulty starting and finishing tasks | Starting | Why am I so paralyzed? Why can't I just do this one thing, it's not that hard it shouldn't be that hard. |
| | | We don't have as much agency over controlling [starting tasks] as easily. Your ability to act is based on things that are just about your chemical physical, what I like to call my meat. My meat doesn't let me do what my thinking wants me to do sometimes. |
| | Finishing | But then there's also many projects that I'm like 60% done, 70% done, I've done the bulk of it but once it gets to documenting what I've done, telling other people what I've done. Anything that seem tedious to me or less interesting, I just stop. It makes it seem like I don't do stuff because I only do things 50% and then no one ever sees the results. Every ADHD person has a graveyard of unfinished projects. |

*Difficulty starting and finishing tasks.* Participants reported struggling tremendously with bringing their attention to starting new things, often feeling trapped by executive dysfunction. Even after starting a task, maintaining attention and ultimately finishing the objective often posed difficulty. This resulted in numerous partially completed tasks with few finished products.

**Emotions.** *Emotional lability*. Intense emotions were commonly described in every focus group, with participants having extreme reactions to stimuli they viewed as minor (Table 4). This was experienced as both emotional highs and lows. Emotions fluctuated quickly with participants reporting going "from zero to one hundred" without gradation. Some participants felt emotions in a binary fashion, either feeling an extremely intense emotion or feeling completely numb. Participants did not describe experiencing emotions in moderation.

Difficulties maintaining responsibilities secondary to inattention and difficulties with task prioritization often generated negative emotions such as frustration and anxiety. One participant viewed their anxiety as an adaptive mechanism to mitigate impulsive spending. Some participants reported difficulty expressing the intensity of their emotions to others, while others reported feeling unable to conceal their reactions.

Although emotional dysregulation was acknowledged by most, three participants stated that they did not experience intense emotions due to their ADHD. One participant stated that their rapid shifts in attention prevented rumination and emotional dysregulation.

*Alexithymia*. In addition to difficulty managing their emotions, participants struggled with identifying and naming their feelings. Difficulties in identifying emotions often generated spirals of frustration and self-blame. Some participants described needing time to experience their emotions; they did not feel them in the moment but rather felt them after they had had time to reflect. This process often led to interpersonal conflicts as they preferred to have distance from an event before reflecting on it with another involved party.

*Rejection-sensitive dysphoria*. Many participants endorsed experiencing rejection-sensitive dysphoria (RSD). Among ADHD communities, RSD is referred to as rejection-sensitivity dysphoria and this term will be used for the rest of this manuscript. Participants described RSD as ruminating over unpleasant emotions, self-blame, and somatization of emotional distress following perceived rejection by others. Common triggers for RSD included feeling excluded from social situations by peers or perceived abandonment from loved ones. Participants described understanding their reactions were often out of proportion to the situations, but they felt unable to control their responses. Other triggers for experiencing RSD included receiving negative feedback regarding work, academic underperformance, and perceived rejection in online spaces.

In response to feeling rejected, many participants retreated from the person by whom they felt rejected and avoided social situations if they feared that they might be excluded. Some participants had developed coping mechanisms for handling their dysphoria, such as reminding themselves not to take things personally in the moment. Learning the term RSD itself was often helpful in understanding the phenomenon and being able to mitigate its effects by making participants cognizant of related tendencies.

Some participants hypothesized why people with ADHD experience RSD. Some viewed it as a learned response to repeated rejection related to their personal struggles with social norms and communicating with others. Others thought that people with ADHD may pick up on social cues that others may not and interpret them as rejection. One participant reported repeatedly putting themselves in relationships where they would be rejected.

Over the course of the focus groups, three participants described never having felt RSD. The same participant who previously described their rapid shifts of attention preventing emotional dysregulation also reflected on their attentional struggles preventing experiencing RSD. They also attributed not experiencing RSD to having strong social support.

**Symptom changes over time.** *Improvement*. A few participants reflected that their symptoms had decreased in intensity since childhood, most notably hyperactivity and sensory sensitivity (Table 5). Far more common was the perception that the symptoms of ADHD had not changed since childhood. Rather, participants felt that they had simply developed better coping

**Table 4. Representative quotes on emotions.**

| Emotional lability | Range of emotions | Emotions can be really difficult to control when it's a particular trigger and it's not something that you can always expect. The way I've always described it is that somebody has lit a match and your nerves and your brain are the fuse and something in there is on fire for a while. |
|---|---|---|
| | | I'm very emotionally volatile. Something will happen and it'll be like this is literally the best day ever or something else will happen and I'll just be so sad that I cannot stop crying for five hours. And it's very little things that set me off. |
| | "All or nothing" | I either feel a lot of emotions very intensely or I feel just kind of, I guess neutral, for lack of a better word. |
| | | I just feel paralyzed like I shut down completely, all the emotions are just gone and I can't do anything anymore. |
| | Negative emotions | I always attribute the fact that I have anxiety to the fact that I think that I developed it as a coping mechanism. I'm going to impulse purchase, but hurry let's be anxious about money, so now we have two problems. |
| | Expressing emotions | My emotions are kind of like, it's like that whole duck thing, swimming along water and they look totally calm but then their feet are going like crazy propelling them along. . . And it's just so bizarre for me because I'm having this really big reaction, and nobody knows until I say something. |
| | | I can't hide my emotions very well. My mom's like you're an open book and I'm like I know. . . I would suck at a game of poker. |
| | Not experiencing | I'm a very even person and I actually do think the ADHD has some kind of effect on that. . . there are a lot of little things that might bother me in the moment but then my attention will switch so quickly to something else that I forget that something was bothering. . . it keeps me from being super up and down on the emotional scale. |
| Alexithymia | | I can't really verbalize what it is that I'm feeling which frustrates me even more because I can't put the words to what I'm feeling . . . If I don't know what's wrong with me then why in the world am I upset? |
| | | It's just like sometimes I don't even know what I feel. I'm like why do I feel these? . . . I don't even know what I feel. |
| | Time to process emotions | I feel like for me with emotions I tend to process it in more of a delayed way because I tend to keep it on the inside and internalize it. So, when something bad happens I just kind of acknowledge that it's there but it doesn't hit me until maybe before I go to sleep and I'm just crying and I just remember everything about my day that's lead up to why I'm crying. |

*(Continued)*

**Table 4.** (Continued)

| Rejection sensitivity dysphoria | Subjective experience | It's literal pain for me, I literally feel it in my chest and it hurts. I'll sit and think about it for an extremely long time and nit-pick it and replay it over and over again to the point where people are like, a lot of time has gone by, it's not that big of a deal, let it go, and I'm still sitting there pondering everything I could have done. |
|---|---|---|
| | Triggers- social exclusion | But if I do randomly hear that my friend is hanging out with someone else for some reason, I feel really insecure for some reason, like I thought you were my friend or something. It doesn't make any sense because of course my friend is going to have other friends. |
| | | But sometimes even if we're joking, I'll ask my boyfriend to go to work with me and obviously he can't so when he tells me no, I just take it so personal, like why is he telling me no? . . .. As soon as the slightest hint of rejection and I'm like it's done, it's over. I already know I'm sensitive and emotional, but this is just a completely different level because it's things that I know, I know he can't go to work with me. |
| | Triggers- negative feedback | In the moment I get completely overwhelmed and I worry about- oh my God am I going to get fired because I messed up this one little thing? . . . It's put me in some embarrassing situations where it shouldn't have been a big deal that I was getting this feedback. |
| | | Yes 100% that exists in me; huge, huge fear of rejection. . .. But rejection I guess from teachers, very much when I didn't meet the grade and wasn't accepted into the advanced class or separated from my peers because I was in the class that needed extra support or something. |
| | Avoidance of social situations | In my experience with rejection, I feel like a lot of the times it can make me self-isolate. If I feel like someone has excluded me over something trivial, I'll just push myself away farther. |
| | Coping | I've been able to look at it and been like okay, there's a reason why that didn't happen so next time let's just think about that instead of automatically getting upset and teaching myself to do that over time has made me not really care as much about it. |
| | | The term rejection sensitivity dysphoria has been helpful in terms of framing. . . and has also helped me visualize the particular issue, and naming something helps you feel like you're in power and control of it. |
| | Ideas on why they experience RSD | I feel like it happens not necessarily as a symptom but as a learned sort of thing. I had a lot of failed relationships and friendships throughout elementary school, and I never really knew why I didn't fit in with a lot of people where we have all these things in common, why don't they like me? And I always felt separated and distant from any friendships and people would kind of become distant and leave for no reason and there was no explanation. So, at that point the only common denominator. . . So, then any time that I get any hint of that happening I freak out and am like oh I'm going to be abandoned again, I'm going to be left alone again. |
| | Not experiencing | Rejection sensitivity dysphoria has been one that I don't think I've ever personally really felt, which I'm kind of grateful for because it does sound pretty terrible for the folks that do feel it. But I'm perfectly happy most of the time to just switch my attention, especially in a social circumstance. I have a good support system and friend group, so if I'm not invited to something I'm not worried about being purposely excluded from it. |

**Table 5. Representative quotes on symptom changes over time.**

| Improvement | Intrinsic | I had a lot more sensory issues as a child. . . I had a lot more issues with noise and stuff than I do now. |
|---|---|---|
| | | I've gotten kind of less hyperactive in like a wiggly sort of way. |
| | Environmental change | I don't know that the symptoms themselves have changed very much since I was a child . . . a lot of my symptoms are probably the same but the way that I deal with them are different. Like a lot of people have already mentioned, I've learned to mask over that time and learned a lot of coping mechanisms over that time that I didn't have when I was younger. I don't tend to have to get new keys cut every other week these days because I always leave them in the same spot. |
| | | Since I've recently learned about my diagnoses and stuff, being able to put a name to it and a reason as to why I am the way I am, it's kind of helped. Instead of fighting it, working with it. |
| | | I don't think the symptoms have really changed for me, but I've just become more aware, and the medication has helped me. I guess I focus a lot better now. |
| Worsening | Intrinsic | As I grow older, I'm not as open minded to learn and pick up new things. |
| | | My memory has gotten a lot worse . . . I don't remember where I put things, I don't remember saying things, sometimes I feel like an old person because my memory has really been my biggest struggle with ADHD in my 30s. |
| | Environmental change | I feel like it has gotten worse, but I feel a lot of that can be attributed to not having the same structure that children have in school. Also as being an adult, you have to deal with everything beyond just your work or school there's just everything you have to do in your life. |
| | | I find it harder as an adult than as a child just because as a child you have so many barriers and people that are kind of supporting you . . . There's a schedule that you're forced to stick to while as an adult there's nothing stopping you . . . I think as a child it's much easier because there's so many rules and things that are outside your control that are keeping you in check. |

skills to manage symptoms. Such strategies included masking of symptoms and increased self-awareness after having been diagnosed as an adult. Stimulant medication was also seen as advantageous in managing symptoms.

*Worsening.* Some participants stated that their ADHD symptoms had become more pronounced in adulthood, such as having increased struggles with memory. More frequently, participants felt that their symptoms had remained the same since childhood, but the lack of structure and support in adult life combined with added stressors made managing their symptoms more difficult.

## Discussion

This qualitative focus group study explores how young adults with ADHD perceive their condition, conceptualize their experiences relative to the current diagnostic criteria, and view attentional and emotional symptoms of ADHD. The sample was largely female, diagnosed with the inattentive sub-type, and highly educated. Having been recruited from online communities for adults with ADHD, participants were likely to be highly invested in advocating for issues surrounding those with ADHD.

These findings on perception of diagnosis support previous work stating that those diagnosed with ADHD in adulthood were often originally misdiagnosed [29] and the phenomenon of adults wishing they had been diagnosed with ADHD sooner [28, 30–35]. Unique to this study are participant perceptions on the diagnostic criteria themselves. Participants largely felt that the current criteria did not describe their experiences and that executive dysfunction may better encompass their symptoms. Renaming of ADHD to executive function disorder has been previously proposed [36]. There was also apprehension with including "developmentally inappropriate" in the description of ADHD, as it was perceived as offensive and not applicable for adults.

The experience of attentional dysregulation described in this study, as opposed to a broad deficit of attention, is consistent with the current body of literature on adult ADHD. Hyperfocusing was first described as a feature of ADHD in 2005 [11], and there have been previous qualitative studies exploring the contextual nature of attention in ADHD. Adults with ADHD

have reported being able to focus easily on tasks of personal interest [37–39] and being able to attend more easily to novel tasks [39, 40]. Difficulty focusing in distracting environments, especially with respect to sounds, has been previously described [40–42]. People with ADHD have reported having difficulty starting assignments until the "last minute" [38, 39, 43] and having difficulty finishing tasks [44]. Attentional symptoms reported in this study which have typically not been previously described include the drawbacks of hyperfocusing, such as neglecting other tasks, experiencing time blindness, and becoming irritable when interrupted while hyperfocusing.

Findings on emotional dysregulation in this study support previous qualitative work. It has been well described that adults with ADHD report intense emotions and emotional lability [32, 33, 37, 38, 40, 41, 45] and often experience unpleasant emotions such as anxiety, depression, and frustration [31, 32, 35, 38, 45, 46]. Alexithymia has been described as affecting 22% of adults with ADHD, although their mean scores on an alexithymia rating scale were not significantly different from non-ADHD controls [47]. It has been proposed that parenting style when growing up and severity of ADHD symptoms mediate alexithymia and emotional processing in adults with ADHD [48]. In one qualitative study of adults with substance use disorders and ADHD, participants reported that they did not recognize or understand their emotions well [45]. Further exploration of the relationships between alexithymia and ADHD is needed.

This is the first qualitative study to our knowledge that specifically explores adults with ADHD's experiences with RSD. Participants in previous qualitative studies on adult ADHD have described experiences that could be interpreted as RSD such as experiencing frequent rejection [35], having a heightened sensitivity to criticism [49], and avoiding others for fear of social problems [38, 45, 50]. Youth with ADHD report higher rejection sensitivity and justice sensitivity on questionnaires [51], although college men with ADHD did not score differently from controls on questionaries for rejection sensitivity [52]. As commented on by participants in this study, RSD may be a learned behavior secondary to being frequently rejected due to struggles with social interactions as opposed to a cardinal feature of ADHD. It should be noted that RSD, although common, was not reported by every participant in this study. Future studies are needed to investigate the prevalence of RSD in adult ADHD, to determine whether demographics underrepresented here such as males and those with the hyperactive sub-type commonly experience RSD, and to elucidate whether it is an intrinsic feature of ADHD or a response to social interactions.

Although ADHD has been reported to persist into adulthood in about half of cases [4], little research has been done on how ADHD symptoms change over the lifespan. One previous study has described that neurocognition worsened in adulthood among 14% of people with ADHD and that participants reported increased demands and less organizational support in adulthood [53]. It has been proposed that ADHD can sometimes worsen in adulthood as the discrepancy between societal expectations and executive function widen, as decreases occur in familial support, and as individuals struggle with additional concerns such as substance use [36]. More research is needed to disentangle the extent to which symptoms of ADHD change over the lifespan due to neurodevelopmental changes or changing societal expectations, and to what extent other factors may contribute.

This study population was predominantly from the United States and Canada; perceptions of ADHD change considerably across countries and geographic regions. Despite prevalence of ADHD not significantly differing between North America and Europe over the past few decades when accounting for methodological differences [2, 54, 55], there is widespread public perception that ADHD is an American condition. For example, a controversial and eventually partially redacted article originally titled, "Why French Kids Don't have ADHD" [56], gained

considerable traction in the early 2010s. Different countries have varied perspectives on ADHD and whether it should be primarily treated pharmacologically or behaviorally [57, 58]. The content of these articles suggest differences in how ADHD is conceptualized across different nations and cultures. Perhaps populations outside North America would have different opinions on the current diagnostic criteria or have different perspectives on attributing additional symptoms to their ADHD. Further work is needed to characterize ADHD symptoms across cultures.

This study has limitations. The sample was predominately White women from the United States who were diagnosed in adulthood with the inattentive ADHD subtype, all of whom had access to technology; thus, there are likely perspectives that were not represented in this study. For example, the current study suggests that adults with ADHD present with executive dysfunction, and more often with dysregulated attention than overarching attention deficits. This may not be the case in adults who were diagnosed with ADHD as children, who may present more with overarching attention deficit than attention dysregulation. Dissatisfaction with current diagnostic criteria was prevalent in the current study, but that may not be the case in men, those with the hyperactive ADHD subtype, or those diagnosed in childhood.

Emotional dysregulation and RSD were more commonly reported by females than males in this study, with one out of seven males and fifteen out of thirty-six females reporting labile emotions and three out of seven males and thirty out of thirty-six females reporting RSD. As this was a focus group study, questions were not systematically asked to every participant to comment on individually, and these results may not fully capture the prevalence of these symptoms. However, the data suggest that women may be more likely to experience emotional dysregulation and RSD, and further work should be done to characterize these symptoms in a more male predominant sample. As this sample was largely White, there were not enough participants of different racial and ethnic groups to make informed comparisons between groups. Further research should investigate ADHD symptom presentation across different races and cultures. Furthermore, it would be important to compare prevalence of the symptoms reported here in comparison to a non-ADHD sample. The study participants were also largely well educated. Future studies should include and investigate groups with varying, including lower, levels of education.

Given the current composition of the study participants, it is possible that people with the aforementioned demographics comprise the majority of members of the online communities from which participants were recruited, people with these demographics were more likely to participate in this study, or some other factors were in operation. The demographics of the communities from which participants were recruited are currently unknown. We hypothesize that because women and those with the inattentive sub-type may be more likely to go undiagnosed until adulthood, they may be more likely to seek out previously missing resources when they ultimately receive a diagnosis.

Although all participants self-reported a formal diagnosis of ADHD and scored above threshold on the Adult ADHD Self-Report Scale, diagnosis was only confirmed in 63% of participants. Reasons for some participants not having a confirmed diagnosis largely consisted of participants verbally agreeing to have their healthcare provider contacted but then the research team not receiving release-of-information documentation or being unable to establish contact with their providers. As all participants verbally agreed to having their healthcare providers contacted, data from all participants regardless of diagnostic confirmation were included in this study for inclusivity.

The sample was highly educated with 88% having some college. This may be because those without higher education were more likely to have limited access to the internet to engage in the study. Further, how participants perceived some of the constructs may have varied. For example, the term executive dysfunction may mean different things to different people, and

further studies investigating specific aspects are warranted. As this study employed focus groups, there is a risk of participant conformity to other members of the group. Furthermore, the use of initially more directed questions may have influenced responses, and future studies may consider the use of more frequent open-ended questions. Additionally, there was a high level of non-response among people who initially completed the screening survey, therefore making this a self-selecting group of people.

## Conclusions

This study explores how young adults with ADHD perceive their condition and experience symptoms, and how these symptoms may change over time. Findings from this predominantly female sample support that attention dysregulation, as opposed to attention deficit, may better describe ADHD among some young adults and that the experience of hyperfocusing may either be helpful or may cause problems. Emotional dysregulation was widely reported, including emotional lability, alexithymia, and RSD. Symptoms improving or worsening over time were most commonly seen to be a reflection of a changing environment as opposed to a change in the symptom presentation themselves. While some of the concepts described in this manuscript have been communicated in various online forums, they have been less well described in the scientific literature. Further, unanticipated findings were observed including ones relating to triggers for RSD, ideas on why young adults with ADHD may experience RSD, irritability upon being interrupted when hyperfocusing, alexithymia, and time blindness. As such, the focus group work illuminated novel understandings and suggests new areas of research. The current findings and those from suggested future studies may assist in refining diagnostic criteria for adults with ADHD.

## Supporting information

**S1 Appendix. Consent meetings script.**
(DOCX)

**S2 Appendix. Informed consent document distributed to participants.**
(DOCX)

**S3 Appendix. Discussion guide used during focus groups.**
(DOCX)

**S4 Appendix. All quotations from the series of focus groups organized thematically.**
(XLSX)

## Acknowledgments

We would like to thank the Facebook and Subreddit moderators and the Children and Adults with ADD (CHADD) advocacy group for their assistance with participant recruitment for this study.

## Author Contributions

**Conceptualization:** Callie M. Ginapp, Gustavo A. Angarita, Krysten W. Bold, Marc N. Potenza.

**Formal analysis:** Callie M. Ginapp, Norman R. Greenberg.

**Funding acquisition:** Callie M. Ginapp.

**Investigation:** Callie M. Ginapp, Grace MacDonald-Gagnon.

**Methodology:** Callie M. Ginapp, Gustavo A. Angarita, Krysten W. Bold, Marc N. Potenza.

**Project administration:** Callie M. Ginapp.

**Supervision:** Gustavo A. Angarita, Krysten W. Bold, Marc N. Potenza.

**Writing – original draft:** Callie M. Ginapp.

**Writing – review & editing:** Norman R. Greenberg, Grace MacDonald-Gagnon, Gustavo A. Angarita, Krysten W. Bold, Marc N. Potenza.

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
