## [Decision Letter · Decision Letter 0]

12 Dec 2022

PONE-D-22-28008“Dysregulated not deficit”: a qualitative study on symptomatology of ADHD in young adultsPLOS ONE

Dear Dr. Ginapp,

Thank you for submitting your manuscript to PLOS ONE. After careful consideration, we feel that it has merit but does not fully meet PLOS ONE’s publication criteria as it currently stands. Therefore, we invite you to submit a revised version of the manuscript that addresses the points raised during the review process. Finding timely reviews has been a challenge recently.  As Academic Editor I am proceeding on the basis of having secured one detailed review (see below). 

We look forward to receiving your revised manuscript.

Kind regards,

Charlotte Lennox

Academic Editor

PLOS ONE

Journal Requirements:

"The authors report no conflicts of interest with the topic of this work. Marc N. Potenza has consulted for and advised Opiant Pharmaceuticals, Idorsia Pharmaceuticals, Baria-Tek, AXA, Game Day Data and the Addiction Policy Forum; has been involved in a patent application with Yale University and Novartis; has received research support from the Mohegan Sun Casino and Connecticut Council on Problem Gambling; has participated in surveys, mailings or telephone consultations related to drug addiction, impulse control disorders or other health topics; and has consulted for law offices and gambling entities on issues related to impulse control or addictive disorders. The other authors report no disclosures."

Reviewers' comments:

Reviewer's Responses to Questions

**Comments to the Author**

1. Is the manuscript technically sound, and do the data support the conclusions?

Reviewer #1: Yes

2. Has the statistical analysis been performed appropriately and rigorously? 

Reviewer #1: N/A

3. Have the authors made all data underlying the findings in their manuscript fully available?

Reviewer #1: Yes

4. Is the manuscript presented in an intelligible fashion and written in standard English?

Reviewer #1: Yes

5. Review Comments to the Author

Reviewer #1: This qualitative manuscript aimed to explore young adults perceptions of ADHD diagnostic criteria. The authors found discord between patients perceptions of ADHD and the diagnostic criteria and conclude that the current diagnostic criteria for ADHD may not best represent the symptoms experienced in adults. The article is well written, it was a pleasure to read and adds an interesting and useful addition to the literature. I make further recommendations for the authors to consider below.

Abstract:

- Please add gender of sample

- Please add country of participants. We know that there’s a considerable difference in the perceptions of ADHD in USA vs Europe for example

Introduction

- Well written and covers the relevant literature on ADHD symptoms and how these may present in adults

- I think it would be useful to briefly cover about missed diagnosis in childhood and then how ADHD is diagnosed in adulthood (e.g. briefly cover typical methods used to diagnose adult ADHD) and the impact that a delayed diagnosis may have

- It may also be useful to briefly mention how gender may influence ADHD diagnosis being missed in childhood, for example, the inattentive female may be more likely to be missed. On that note, it is interesting that your sample is predominately female and also did not agree with hyperactivity being a core symptom of ADHD – worthy of discussion

Method

- Please state country where participants were recruited/research conducted

- It’s a real strength that you asked for confirmation of diagnosis and reaching a cut-off on a symptom scale. Please state why you chose not to exclude those who didn’t provide this and did you explore whether answers differed accordingly.

Results

- I would like to have seen some comparison in the results exploring whether there were any differences between males/females and also different ethnicities. We know that these can be very important factors in interpreting ADHD symptoms, one which does require more research, it feels a missed opportunity not to qualitatively explore this in your sample. I appreciate you would not want to make any firm conclusions on this, but a reflection and comparison would add to this paper.

- The quotes were well chosen and reflected the points accurately

Discussion

- Please reflect on how your findings may reflect the country of the sample – for example, would you expect different findings in Europe and why

- Please reflect on the fact that 93% of your sample are likely to be quite educated/advocates on ADHD (being part of social media groups)

- It is good to see here the discussion between women inattentive and males and hyperactive subtype – please bring this in earlier (intro/results).

- In the discussion, is it worth reflecting that perhaps the sample are also not sure what are ADHD characteristics and what are characteristics of other co-morbid disorders or general difficulties that a non-clinical sample of young adults may also experience. For example, it may be common young adults starting work may also experience rejection-sensitive dysphoria. Your sample is predominately female and emotional dysregulation is more prominent in females than males regardless of ADHD. This requires discussion, and if you feel appropriate, the conclusion could be tempered. It is hard to conclude one directional line in a qualitative study, particularly where you did not also interview non-ADHD sample.

6. PLOS authors have the option to publish the peer review history of their article (what does this mean?). If published, this will include your full peer review and any attached files.

Reviewer #1: No

---

## [Author Response · Author response to Decision Letter 0]

16 Dec 2022

 The manuscript has been formatted according to the style requirements. 

"The authors report no conflicts of interest with the topic of this work. Marc N. Potenza has consulted for and advised Opiant Pharmaceuticals, Idorsia Pharmaceuticals, Baria-Tek, AXA, Game Day Data and the Addiction Policy Forum; has been involved in a patent application with Yale University and Novartis; has received research support from the Mohegan Sun Casino and Connecticut Council on Problem Gambling; has participated in surveys, mailings or telephone consultations related to drug addiction, impulse control disorders or other health topics; and has consulted for law offices and gambling entities on issues related to impulse control or addictive disorders. The other authors report no disclosures."

The statement has been added to the competing interests and the competing interests has been added to the cover letter. 

The full name of the IRB has been included and it now states we obtained verbal consent from individuals who participated in this study. 

The caption for the supporting information is now included in the text. 

Reviewers' comments:

Reviewer's Responses to Questions

Comments to the Author

1. Is the manuscript technically sound, and do the data support the conclusions?

Reviewer #1: Yes

2. Has the statistical analysis been performed appropriately and rigorously? 

Reviewer #1: N/A

3. Have the authors made all data underlying the findings in their manuscript fully available?

Reviewer #1: Yes

4. Is the manuscript presented in an intelligible fashion and written in standard English?

Reviewer #1: Yes

5. Review Comments to the Author

Reviewer #1: This qualitative manuscript aimed to explore young adults perceptions of ADHD diagnostic criteria. The authors found discord between patients perceptions of ADHD and the diagnostic criteria and conclude that the current diagnostic criteria for ADHD may not best represent the symptoms experienced in adults. The article is well written, it was a pleasure to read and adds an interesting and useful addition to the literature. I make further recommendations for the authors to consider below.

Abstract:

- Please add gender of sample

- Please add country of participants. We know that there’s a considerable difference in the perceptions of ADHD in USA vs Europe for example

Response: Participant sex and continent have been added to the abstract. 

Introduction

- Well written and covers the relevant literature on ADHD symptoms and how these may present in adults

- I think it would be useful to briefly cover about missed diagnosis in childhood and then how ADHD is diagnosed in adulthood (e.g. briefly cover typical methods used to diagnose adult ADHD) and the impact that a delayed diagnosis may have

Response: It is now covered in the introduction that ADHD is underdiagnosed and there are often substantial delays in diagnosis. A brief overview of methods of diagnosis in adulthood has been added along with the potential impacts of a delayed diagnosis. 

- It may also be useful to briefly mention how gender may influence ADHD diagnosis being missed in childhood, for example, the inattentive female may be more likely to be missed. On that note, it is interesting that your sample is predominately female and also did not agree with hyperactivity being a core symptom of ADHD – worthy of discussion

Response: It is now included that females and those with the inattentive subtype are underdiagnosed with ADHD and that our sample is predominantly female. 

Method

- Please state country where participants were recruited/research conducted

Response: It has been stated that this study was based out of the United States. The full list of countries where participants were located has been added. 

- It’s a real strength that you asked for confirmation of diagnosis and reaching a cut-off on a symptom scale. Please state why you chose not to exclude those who didn’t provide this and did you explore whether answers differed accordingly.

Response: It is now included in the limitations that diagnosis was only confirmed in 63% of participants. As reasons for not confirming diagnosis were largely participants verbally agreeing to have their provider contacted but then the research team not receiving release-of-information documentation or being unable to establish contact with their providers, data from all participants were included in the study. It was not explored whether answers differed accordingly. 

Results

- I would like to have seen some comparison in the results exploring whether there were any differences between males/females and also different ethnicities. We know that these can be very important factors in interpreting ADHD symptoms, one which does require more research, it feels a missed opportunity not to qualitatively explore this in your sample. I appreciate you would not want to make any firm conclusions on this, but a reflection and comparison would add to this paper.

Response: Thank you for bringing up this important point. Differences in number of participants reporting emotional dysregulation and rejection-sensitive dysphoria are now included in the limitations, along with comment that these proportions may not be representative of overall prevalence of these symptoms. As the study was predominantly White, there were not enough participants of other racial and ethnic groups to make informed comparisons between groups. We now note that this is a limitation that should be examined in future studies. 

- The quotes were well chosen and reflected the points accurately

Response: Thank you for this comment. 

Discussion

- Please reflect on how your findings may reflect the country of the sample – for example, would you expect different findings in Europe and why

Response: A paragraph has been added to the discussion detailing different perspectives on ADHD across cultures, particularly the US and Europe. For example, despite there not being differences in prevalence of ADHD between the US and Europe, there is widespread public perception that ADHD is an American condition and that may influence how participants view diagnostic criteria or conceptualize their symptoms. 

- Please reflect on the fact that 93% of your sample are likely to be quite educated/advocates on ADHD (being part of social media groups)

Response: It is now included at the beginning of the discussion that the sample was recruited from online communities for adults with ADHD and are likely to be highly invested in advocating for issues surrounding those with ADHD. We also note the highly educated aspect of the sample and cite the need for study of other populations with other levels of education. 

- It is good to see here the discussion between women inattentive and males and hyperactive subtype – please bring this in earlier (intro/results).

Response: The higher prevalence of the inattentive subtype in women has now been added to the introduction. 

- In the discussion, is it worth reflecting that perhaps the sample are also not sure what are ADHD characteristics and what are characteristics of other co-morbid disorders or general difficulties that a non-clinical sample of young adults may also experience. For example, it may be common young adults starting work may also experience rejection-sensitive dysphoria. Your sample is predominately female and emotional dysregulation is more prominent in females than males regardless of ADHD. This requires discussion, and if you feel appropriate, the conclusion could be tempered. It is hard to conclude one directional line in a qualitative study, particularly where you did not also interview non-ADHD sample.

Response: It is now further discussed in the limitations that the sample was largely female and that this feature may contribute to the findings of emotional dysregulation and rejection-sensitive dysphoria. Further areas for research including comparing prevalence of these symptoms across demographic features within those with ADHD and in comparison to those without ADHD are now included.

---

## [Decision Letter · Decision Letter 1]

15 Mar 2023

PONE-D-22-28008R1“Dysregulated not deficit”: a qualitative study on symptomatology of ADHD in young adultsPLOS ONE

Dear Dr. Ginapp,

Thank you for submitting your manuscript to PLOS ONE. After careful consideration, we feel that it has merit but does not fully meet PLOS ONE’s publication criteria as it currently stands. Therefore, we invite you to submit a revised version of the manuscript that addresses the points raised during the review process.

We look forward to receiving your revised manuscript.

Kind regards,

Nabeel Al-Yateem, PhD

Academic Editor

PLOS ONE

Reviewers' comments:

Reviewer's Responses to Questions

**Comments to the Author**

1. If the authors have adequately addressed your comments raised in a previous round of review and you feel that this manuscript is now acceptable for publication, you may indicate that here to bypass the “Comments to the Author” section, enter your conflict of interest statement in the “Confidential to Editor” section, and submit your "Accept" recommendation.

Reviewer #2: (No Response)

Reviewer #3: (No Response)

2. Is the manuscript technically sound, and do the data support the conclusions?

Reviewer #2: No

Reviewer #3: Yes

3. Has the statistical analysis been performed appropriately and rigorously? 

Reviewer #2: N/A

Reviewer #3: Yes

4. Have the authors made all data underlying the findings in their manuscript fully available?

Reviewer #2: No

Reviewer #3: Yes

5. Is the manuscript presented in an intelligible fashion and written in standard English?

Reviewer #2: Yes

Reviewer #3: Yes

6. Review Comments to the Author

Reviewer #2: This is a qualitative study, investigating the perception of the relevance of the diagnostic criteria for ADHD, through focus groups, using the IPA method to analyse the verbatim of the exchanges in the groups.

Forty-three people, mostly women, self-reporting a diagnosis of ADHD, were recruited from online communities around ADHD.

The results suggest a mismatch in the wording and content of the DSM diagnostic criteria, and highlight that certain notions are prevalent in the participants' perception of symptoms, such as rejection sensitivity dysphoria and hyperfocus.

This study has the interest of highlighting the limits of the diagnostic criteria for ADHD in young adults, but I feel that it does not add much to current knowledge, with a rather low level of evidence.

My main criticism is that these notions (RSD and hyperfocus) are widely discussed in forums, websites and social networks about ADHD, and have been for several years, in the United States and elsewhere, long before this study was conducted. It seems fairly obvious to me that there is a reproduction of a fairly standard discourse, and few new hypotheses obtained.

The authors could have mentioned, or possibly evaluated, the frequency of these data in specialized websites and social groups around ADHD.

It seems to me that this is the main interest of a qualitative study to generate new ideas, and I feel that this study does not really succeed in doing so.

Otherwise, the manuscript is clearly written.

My main criticism is particularly illustrated by the content of the introduction. The authors refer to and argue for the hypotheses they draw from their qualitative work, which is a bit surprising: one gets the impression that they lack the openness to all the hypotheses that the IPA methodology recommends. All in all, I fear that the use of the qualitative IPA method will lead to something rather tautological with this approach.

Other Major points

2/ In the introduction, some points are missing such as general explanations on the qualitative method and its interest in this context, discussion on the genesis of the ADHD criteria in the current classifications.

3/ Method

I am quite surprised to see an IPA study using focus groups.

I am not aware if it has been done before, and the theoretical and practical limitations and problems of this approach should be further discussed.

In the manuscript, this point is only very briefly addressed: (p. 8, l.185 : “Although IPA traditionally utilizes individual interviews (23), focus groups were employed in this study”)

Why did the IPA method was chosen? I have the impression that the analysis is essentially thematic.

I wonder whether using group interviews does not also increase the risk of conformity to a standard discourse.

In addition, it would be useful to mention whether all participants took part in the focus groups, I don't think I have read this information.

4/ The interview grid (its final version) should be offered to the reader.

5/ I appreciated the discussion of the cultural limitations of the study, but I think that limiting, or at least centring this discussion around the 53 reference is problematic. Being French myself, I re-read the quoted article, and it is truly laughable, so full of grotesque clichés about child-rearing in France. Especially since the prevalence of ADHD is about the same as in many other countries, and the low prescription is mainly due to a dominant psychoanalytical model. The current sharp increase in prescribing in France is simply a sign that this model is gradually becoming outdated.

Minor points

p.4 : I am not sure that the prevalence of 7% in adults is the more consensual data.

Reviewer #3: In their submitted paper, Ginapp and colleagues present data from focus groups involving young adults with a diagnosis or symptoms of ADHD. They recruited a total of 43 subjects from online sources who reported being diagnosed with ADHD by a physician and scoring 23 or higher on the ADHD Self-Report Scale for Adults. All subjects completed a screening questionnaire, and if they met the inclusion criteria, they were asked to participate in video- and audio-recorded focus groups that took place over Zoom with participant consent. The recordings were transcribed verbatim, and the transcripts were analyzed using Interpretative Phenomenological Analysis (IPA). The authors examined how participants perceived the official diagnostic criteria. They found that most participants reported that the diagnostic criteria did not accurately reflect their own experiences with ADHD symptoms. Some study participants suggested that executive dysfunction might better capture their symptoms. In addition, participants reported symptom changes over time in the categories of emotional dysregulation, emotional lability, alexithymia, and rejection-sensitive dysphoria (RSD).

The manuscript is well written and the methodology of the study is sound. The results of this study undoubtedly have clinical implications and may be relevant to refining diagnostic criteria.

My only recommendation is that the authors consider rewording the following sentence, "One participant indicated that their rapid shifts in attention prevented rumination and emotional dysregulation." Please rephrase to: "...that his/her rapid attention shifts...". There are numerous sentences throughout the manuscript with a similar structure.

7. PLOS authors have the option to publish the peer review history of their article (what does this mean?). If published, this will include your full peer review and any attached files.

Reviewer #2: No

Reviewer #3: No

---

## [Author Response · Author response to Decision Letter 1]

29 Mar 2023

Reviewer #2: This is a qualitative study, investigating the perception of the relevance of the diagnostic criteria for ADHD, through focus groups, using the IPA method to analyse the verbatim of the exchanges in the groups.

Forty-three people, mostly women, self-reporting a diagnosis of ADHD, were recruited from online communities around ADHD.

The results suggest a mismatch in the wording and content of the DSM diagnostic criteria, and highlight that certain notions are prevalent in the participants' perception of symptoms, such as rejection sensitivity dysphoria and hyperfocus.

This study has the interest of highlighting the limits of the diagnostic criteria for ADHD in young adults, but I feel that it does not add much to current knowledge, with a rather low level of evidence.

My main criticism is that these notions (RSD and hyperfocus) are widely discussed in forums, websites and social networks about ADHD, and have been for several years, in the United States and elsewhere, long before this study was conducted. It seems fairly obvious to me that there is a reproduction of a fairly standard discourse, and few new hypotheses obtained.

The authors could have mentioned, or possibly evaluated, the frequency of these data in specialized websites and social groups around ADHD.

It seems to me that this is the main interest of a qualitative study to generate new ideas, and I feel that this study does not really succeed in doing so.

Otherwise, the manuscript is clearly written.

My main criticism is particularly illustrated by the content of the introduction. The authors refer to and argue for the hypotheses they draw from their qualitative work, which is a bit surprising: one gets the impression that they lack the openness to all the hypotheses that the IPA methodology recommends. All in all, I fear that the use of the qualitative IPA method will lead to something rather tautological with this approach.

Thank you for these points. Analyzing content of social media posts was considered when conceptualizing this project. However, it was decided not to take this approach due to concerns about using these data for research purposes. Many platforms for adult ADHD explicitly state that no content should be used for research. Even among platforms that do not comment on research usage, the people posting the content are not doing so under the impression that it will be utilized in this manner. Because of this, it was decided to sample people from online communities where adult ADHD is discussed so that participants could engage in the study fully aware that their words would be used for research. Although topics such as rejection sensitivity dysphoria (RSD) and hyperfocusing may have been described for years on social media, they are largely lacking in the published peer-reviewed literature. This study sought to bridge that gap. Questions on RSD and hyperfocusing were directly asked in the focus groups, but the data analysis took an inductive approach, and many topics emerged from the focus groups that we did not expect to find. These include triggers for RSD, ideas on why young adults with ADHD may experience RSD, alexithymia, time blindness, and irritability upon being interrupted when engaged in hyperfocusing. We have added some of this material to the manuscript.

Other Major points

2/ In the introduction, some points are missing such as general explanations on the qualitative method and its interest in this context, discussion on the genesis of the ADHD criteria in the current classifications.

A sentence generally explaining the qualitative method has been added to the introduction. There is also now more context around the history of ADHD. 

3/ Method

I am quite surprised to see an IPA study using focus groups.

I am not aware if it has been done before, and the theoretical and practical limitations and problems of this approach should be further discussed.

In the manuscript, this point is only very briefly addressed: (p. 8, l.185 : “Although IPA traditionally utilizes individual interviews (23), focus groups were employed in this study”)

Why did the IPA method was chosen? I have the impression that the analysis is essentially thematic.

An IPA approach adapted for focus groups was chosen because of the focus on participants’ subjective lived experiences and how people ascribe meaning to these lived experiences. We performed analysis on the individual level before performing wider group analysis, consistent with Love et al’s paper detailing how an IPA approach can be modified for analysis of focus groups. More detail has been added to the data analysis and theoretical framework subsections. 

Love, B., Vetere, A., & Davis, P. (2020). Should Interpretative Phenomenological Analysis (IPA) be Used With Focus Groups? Navigating the Bumpy Road of “Iterative Loops,” Idiographic Journeys, and “Phenomenological Bridges”. International Journal of Qualitative Methods, 19, 1609406920921600

I wonder whether using group interviews does not also increase the risk of conformity to a standard discourse.

This has been added to the limitations section. 

In addition, it would be useful to mention whether all participants took part in the focus groups, I don't think I have read this information.

It has been clarified under the procedure section of the methods that all 43 participants took part in a focus group. 

4/ The interview grid (its final version) should be offered to the reader.

The discussion guide has been attached as an appendix. 

5/ I appreciated the discussion of the cultural limitations of the study, but I think that limiting, or at least centering this discussion around the 53 reference is problematic. Being French myself, I re-read the quoted article, and it is truly laughable, so full of grotesque clichés about child-rearing in France. Especially since the prevalence of ADHD is about the same as in many other countries, and the low prescription is mainly due to a dominant psychoanalytical model. The current sharp increase in prescribing in France is simply a sign that this model is gradually becoming outdated.

We regret having initially centered the point around cultural differences in ADHD perceptions around this singular article. Additional citations have been added to communicate a broader view of cultural differences.

Minor points

p.4 : I am not sure that the prevalence of 7% in adults is the more consensual data.

It has been clarified that estimates are between 3%-7%. 

Reviewer #3: In their submitted paper, Ginapp and colleagues present data from focus groups involving young adults with a diagnosis or symptoms of ADHD. They recruited a total of 43 subjects from online sources who reported being diagnosed with ADHD by a physician and scoring 23 or higher on the ADHD Self-Report Scale for Adults. All subjects completed a screening questionnaire, and if they met the inclusion criteria, they were asked to participate in video- and audio-recorded focus groups that took place over Zoom with participant consent. The recordings were transcribed verbatim, and the transcripts were analyzed using Interpretative Phenomenological Analysis (IPA). The authors examined how participants perceived the official diagnostic criteria. They found that most participants reported that the diagnostic criteria did not accurately reflect their own experiences with ADHD symptoms. Some study participants suggested that executive dysfunction might better capture their symptoms. In addition, participants reported symptom changes over time in the categories of emotional dysregulation, emotional lability, alexithymia, and rejection-sensitive dysphoria (RSD).

The manuscript is well written and the methodology of the study is sound. The results of this study undoubtedly have clinical implications and may be relevant to refining diagnostic criteria.

My only recommendation is that the authors consider rewording the following sentence, "One participant indicated that their rapid shifts in attention prevented rumination and emotional dysregulation." Please rephrase to: "...that his/her rapid attention shifts...". There are numerous sentences throughout the manuscript with a similar structure.

Thank you so much for your reflections on this manuscript. 

The use of the singular “they” was employed to be inclusive of pronouns beyond he and she. This was considered especially prudent as 16% of the sample reported being non-binary. We have now communicated this more clearly in the manuscript.

---

## [Decision Letter · Decision Letter 2]

2 Jul 2023

PONE-D-22-28008R2“Dysregulated not deficit”: a qualitative study on symptomatology of ADHD in young adultsPLOS ONE

Dear Dr. Ginapp,

Thank you for submitting your manuscript to PLOS ONE. After careful consideration, we feel that it has merit but does not fully meet PLOS ONE’s publication criteria as it currently stands. Therefore, we invite you to submit a revised version of the manuscript that addresses the points raised during the review process.

We look forward to receiving your revised manuscript.

Kind regards,

Nabeel Al-Yateem, PhD

Academic Editor

PLOS ONE

Journal Requirements:

Reviewers' comments:

Reviewer's Responses to Questions

**Comments to the Author**

1. If the authors have adequately addressed your comments raised in a previous round of review and you feel that this manuscript is now acceptable for publication, you may indicate that here to bypass the “Comments to the Author” section, enter your conflict of interest statement in the “Confidential to Editor” section, and submit your "Accept" recommendation.

Reviewer #2: All comments have been addressed

Reviewer #4: All comments have been addressed

2. Is the manuscript technically sound, and do the data support the conclusions?

Reviewer #2: Yes

Reviewer #4: Yes

3. Has the statistical analysis been performed appropriately and rigorously? 

Reviewer #2: I Don't Know

Reviewer #4: Yes

4. Have the authors made all data underlying the findings in their manuscript fully available?

Reviewer #2: Yes

Reviewer #4: Yes

5. Is the manuscript presented in an intelligible fashion and written in standard English?

Reviewer #2: Yes

Reviewer #4: Yes

6. Review Comments to the Author

Reviewer #2: I have carefully re-read the changes made to the manuscript, and for most of the points raised I have received an appropriate response.

I am grateful to the authors for providing the interview grid. I have participated in the conceptualisation and implementation of several studies using IPA, but I do not consider myself a specialist in this method. I am, however, surprised by this grid, which I would not have validated in its current state for a study in which I would have participated. A large proportion of the questions are closed, and in particular half of the questions opening on a subject. This goes against the general spirit of the IPA and I'm a bit worried about strict compliance with the methodology (in addition to my doubts about group analysis).

Knowing the high level of methodological rigour expected by the journal, I think that a reviewer specialising in IPA should give his opinion.

In the absence of this, my doubts do not allow me to support the publication.

Reviewer #4: I think the authors have responded well to the previous comments. My only comment is around the use of verbal consent and how informed consent can be guaranteed. Please could the authors include the specifics of what participants were informed in the method section and a justification as to why written consent could not be obtained. A copy of the participant information sheet and a copy of the verbal script used, should be submitted as additional information.

7. PLOS authors have the option to publish the peer review history of their article (what does this mean?). If published, this will include your full peer review and any attached files.

Reviewer #2: No

Reviewer #4: **Yes: **Charlotte Lennox

---

## [Author Response · Author response to Decision Letter 2]

18 Jul 2023

Reviewer #2: I have carefully re-read the changes made to the manuscript, and for most of the points raised I have received an appropriate response.

I am grateful to the authors for providing the interview grid. I have participated in the conceptualisation and implementation of several studies using IPA, but I do not consider myself a specialist in this method. I am, however, surprised by this grid, which I would not have validated in its current state for a study in which I would have participated. A large proportion of the questions are closed, and in particular half of the questions opening on a subject. This goes against the general spirit of the IPA and I'm a bit worried about strict compliance with the methodology (in addition to my doubts about group analysis).

Knowing the high level of methodological rigour expected by the journal, I think that a reviewer specialising in IPA should give his opinion.

In the absence of this, my doubts do not allow me to support the publication.

Thank you for raising concerns around the discussion guide. Initial directed questions were asked to gauge agreement or disagreement among the group and to understand whether participants had specific experiences. Subsequently, open-ended follow-up questions were asked to better understand participant experiences. This information has been added to the procedure section of the methods. Additionally, the potential that structuring questions in this way influenced the results has been added as a limitation. 

Reviewer #4: I think the authors have responded well to the previous comments. My only comment is around the use of verbal consent and how informed consent can be guaranteed. Please could the authors include the specifics of what participants were informed in the method section and a justification as to why written consent could not be obtained. A copy of the participant information sheet and a copy of the verbal script used, should be submitted as additional information.

Thank you for this comment. Verbal consent was utilized and approved by the IRB due to focus groups taking place virtually. It was seen as unnecessarily burdensome to have participants sign the document remotely and return it. This has been added to the procedure section of the methods. The participant information sheet and script for the consent meeting have been attached as supporting documents.

---

## [Decision Letter · Decision Letter 3]

28 Sep 2023

“Dysregulated not deficit”: a qualitative study on symptomatology of ADHD in young adults

PONE-D-22-28008R3

Dear Dr. Ginapp,

We’re pleased to inform you that your manuscript has been judged scientifically suitable for publication and will be formally accepted for publication once it meets all outstanding technical requirements.

Kind regards,

Nabeel Al-Yateem, PhD

Academic Editor

PLOS ONE

Additional Editor Comments (optional):

Reviewers' comments:

Reviewer's Responses to Questions

**Comments to the Author**

1. If the authors have adequately addressed your comments raised in a previous round of review and you feel that this manuscript is now acceptable for publication, you may indicate that here to bypass the “Comments to the Author” section, enter your conflict of interest statement in the “Confidential to Editor” section, and submit your "Accept" recommendation.

Reviewer #4: All comments have been addressed

2. Is the manuscript technically sound, and do the data support the conclusions?

Reviewer #4: Yes

3. Has the statistical analysis been performed appropriately and rigorously? 

Reviewer #4: Yes

4. Have the authors made all data underlying the findings in their manuscript fully available?

Reviewer #4: Yes

5. Is the manuscript presented in an intelligible fashion and written in standard English?

Reviewer #4: Yes

6. Review Comments to the Author

Reviewer #4: (No Response)

7. PLOS authors have the option to publish the peer review history of their article (what does this mean?). If published, this will include your full peer review and any attached files.

Reviewer #4: **Yes: **Charlotte Lennox

---

## [Editor Report · Acceptance letter]

2 Oct 2023

PONE-D-22-28008R3 

“Dysregulated not deficit”: a qualitative study on symptomatology of ADHD in young adults 

Dear Dr. Ginapp:

I'm pleased to inform you that your manuscript has been deemed suitable for publication in PLOS ONE. Congratulations! Your manuscript is now with our production department. 

Kind regards, 

on behalf of

Dr. Nabeel Al-Yateem 

Academic Editor

PLOS ONE